# Unleashing the Potential of Classification with Semantic Similarity for Deep Imbalanced Regression

## Abstract

Recent studies have empirically demonstrated the feasibility of directly incorporating classification regularizers into Deep Imbalanced Regression (DIR). By segmenting the entire dataset into distinct groups and performing classification regularization on these groups, previous works primarily focused on maintaining the ordinal consistency between the feature space and the label space to capture the continuity of data in DIR. However, this direct integration would also lead the model to focus merely on learning discriminative features during representation learning and potentially distort the geometrical structure of the feature space due to the label imbalance during the fine-tuning phase in DIR. As a result, semantic similarity, namely, instances with similar labels would also be close to each other, can be leveraged to address the imbalance in DIR but has always been ignored. Consequently, the effectiveness of these classification-based approaches would be significantly undermined in DIR. To tackle this problem, we investigate the similarity characteristics of the data in DIR and propose an end-to-end solution to unleash the potential of classification in helping DIR. Specifically, we first split the objective of DIR into a combination of a global inter-group imbalance group classification task and a local intra-group imbalance instance regression task. To fully exploit the potential of classification under the DIR task, we propose both a symmetric and asymmetric soft labeling strategy to capture the global semantic similarity to handle the cross-group imbalance. In the meantime, we employ label distribution smoothing to leverage the instance semantic similarity in addressing the intra-group instance imbalance with a multi-head regressor. Furthermore, we link up the group classification to guide the learning of the multi-head regressor, which can further harness the classification to help the DIR from end to end. Extensive experiments in real-world datasets also validate the effectiveness of our proposed method. The code can be found in `https://anonymous.4open.science/r/ICLR2025submission-9415/README.md`.

## 1 Introduction

Deep imbalanced regression (DIR) aims to perform regression tasks with deep neural networks on particular datasets where certain labels are much less observed than others Yang et al. (2021). While the goal of classification tasks is to predict discrete class labels to model the training distribution, in contrast, the label space in regression is always continuous and infinite. To tackle this problem, recent research focused on capturing the label continuity in DIR.

By segmenting the whole dataset into distinct and continuous groups, previous works incorporated classification regularizers (e.g. representation learning regularizers) to maintain the continuity of the labels in the feature space. Specifically, these researches has explored extensively in preserving the ordinal nature of the feature space. For instance, Gong et al. (2022) proposed a ranking regularization to align the sorting of features with their corresponding labels. Zhang et al. (2023a) used an ordinal entropy regularizer to maintain the ordinal relationships between the feature and the label. Zha et al. (2023) proposed a contrastive regularization to learn both ordinal and discriminative feature representation. Similarly, Keramati et al. (2024) introduced a novel pair selection strategy into

contrastive learning to pull the positive pairs together and push away the negative pairs given their corresponding label distance.

However, these methods primarily focus on learning the ordinal characteristics of the data in the feature space. In the meantime, the integration of classification regularizers would lead the model to concentrate solely on learning discriminative features Zha et al. (2023); Keramati et al. (2024) in representation learning and potentially alter the geometrical structure of the feature space due to the label imbalance during the fine-tuning phase in DIR (e.g. in Fig.1). Therefore, semantic similarity, another aspect of the continuity in DIR where the similarity across labels would also reflect the similarity of their features, is always overlooked by the previous works. For example, in the age regression task, images of age 20 would have similar features to those of ages 15 and 25. Consequently, the knowledge learned from age 20 can be leveraged to approach the age of 25 or 15 if either of them is less observed in training.

Nevertheless, direct incorporation of the classification regularizers would obliterate this semantic similarity (e.g. push away effect of the feature representationsKeramati et al. (2024)), which limits the feasibility of leveraging semantic similarity for tackling the DIR problem. Additionally, these previous works often treated DIR as merely classification tasks. As the label boundaries in regression tasks become more fine-grained (with smaller bin sizes Yang et al. (2021)), these solutions would inevitably lead to a heavy computational burden and eventually become infeasible for DIR.

In this paper, we investigate the semantic similarity in DIR to exploit the potential of classification in helping DIR. Instead of directly incorporating classification regularizers in the feature space as that of previous works Zhang et al. (2023a); Zha et al. (2023); Keramati et al. (2024), we propose an end-to-end solution that tackles the DIR in the combination of 1) a global inter-group imbalance group classification task and 2) a local intra-group imbalance instance (data sample) regression task. We leverage the semantic similarity from both global and local perspectives to unleash the potential of classification to address the imbalance in global and local respectively.

Specifically, we first propose a symmetric descending soft labeling strategy to capture the semantic similarity across the groups in the group classification task. Meanwhile, considering the imbalance across the groups, we also propose an asymmetric soft labeling strategy that incorporates the imbalance priors of the groups into the symmetric soft labeling to tackle the global imbalance classification. These soft labeling strategies leverage the semantic similarity between the groups to tackle the imbalance across the groups, which can effectively capture the intrinsic characteristics of the data in DIR from a global perspective.

Furthermore, we associate the group predictions from the group classification with a multi-head regressor to guide each instance forwarding to its corresponding regressor head in an end-to-end manner. Additionally, to address the imbalance between the instances in each group, we introduce the local label distribution smoothing to capture the intra-group semantic similarity for each instance from a local perspective. Hereby, we unleash the potential of the classification in helping DIR by leveraging the semantic similarity from global to local. We also conduct comprehensive experiments over three real-world DIR benchmarks to validate the effectiveness of our proposed method.

In summary, our contribution can be concluded as the following:

- We divide the objective of DIR into the combination of 1) a global group imbalance classification task and 2) a local instance imbalance regression task.
- We leverage the semantic similarity to unleash the potential of classification in helping regression by proposing a symmetric and asymmetric descending soft labeling strategy and introducing label distribution smoothing to tackle the imbalance from global to local.
- We associate the global group classification with the local instance regression to address the DIR from end to end.

## 2 MOTIVATION

### 2.1 PRELIMINARY

We denote the training set as $\{x_i, y_i\}_{i=1}^N$ where $x_i \in \mathcal{X}, \mathbb{X} \in \mathcal{R}^d$ is the input and $y_i \in \mathcal{Y}, \mathcal{Y} \in \mathbb{R}$ is the label, $d$ is the dimension. As Pintea et al. (2023), we divide the whole dataset into $G$ disjoint but

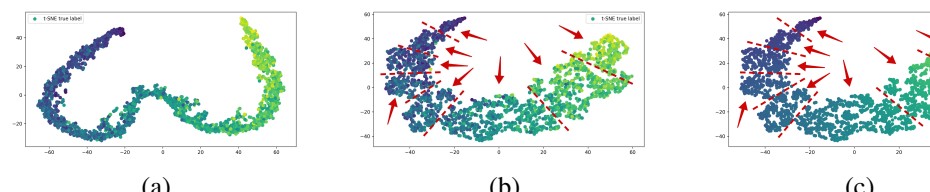

(a)  (b)  (c)

Figure 1: The t-SNE (AgeDB-DIR) of the features in (a) direct classification-based method Zha et al. (2023), (b) classification-based method after fine-tuning (ground truth labeled), (c) classification-based method after fine-tuning (prediction labeled after fine-tuning). We can observe a clear clustering structure in the feature space after fine tuning on the regression tasks on both (b) and (c), which motivates us to exploit the potential of the classification in helping DIR.

continuous groups, each input would correspond to a unique group $g \in \{G\}$ where $\{G\}$ is the set of groups with length $G$, e.g. $\{G\} = \{1, \dots, G\}$ [1]. Also, we denote the deep neural network as the combination of $\{f(\theta), c(\psi), r(\phi^{\{G\}})\}$, where $\theta$ is the parameter of feature extractor $f$, $\phi^{(\cdot)}$ is the parameter of a multi-head regressor $r$. For one arbitrary input $(x, y, g)$, the feature of one arbitrary input $x$ is denoted as $z = f(x, \theta)$, the predicted group of $x$ is denoted as $o_{pred} = c(z, \psi)$ and the predicted label of $x$ is denoted as $y_{pred} = r(z, \phi^g)$ at the head $g$ of regressor $r$. The empirical label density is denoted as $p(y)$.

## 2.2 DISCUSSION ON DIRECT INCORPORATION OF CLASSIFICATION FOR DIR.

Inspired by Pintea et al. (2023), we decompose the objective of DIR into the combination of the group classification globally over the whole dataset and instance regression locally within each group from a Bayesian perspective: $p(y|x) = \sum_g^G p(g|x)p(y|x, g)$, where $g$ denotes the group index and $G$ denotes the total number of groups. Therefore, we can model the $p(g|x)$ as the global group classification and $p(y|x, g)$ as the local instance regression. Consequently, the imbalance of our regression task has also been divided into the global inter-group classification imbalance and local intra-group instance imbalance.

When we take the negative log-likelihood of the objective of DIR, we can have the following decomposition: $-\log p(y|x) = \sum_g^G -\log p(g|x) - \log p(y|x, g)$, where the $-\log p(g|x)$ can be regarded as the group classification loss and $-\log p(y|x, g)$ can be regarded as the regression loss given $g$. As most of the previous works Zha et al. (2023); Zhang et al. (2023a) which incorporated the classification regularizers in the feature space are actually modeling the posterior of the feature representation $p(z|x)$, there exists a gap between modeling the $p(z|x)$ and the $p(g|x)$ in our decomposition.

Furthermore, at the fine-tuning phase Zha et al. (2023), the data dependence Yang et al. (2021) and the label imbalance would also affect the mapping process (from $p(z|x) \rightarrow p(y|z)$ in DIR). Consequently, the geometrical structure of the feature space would be distorted this fine-tuning phase. As we can observe from Fig.1, the structure of the feature space in (a) has been modified by the fine-tuning phase and differs a lot compared to (a) in both (b) and (c) given the label imbalance in DIR. Therefore, the effectiveness of incorporating classification regularizers would be significantly limited in addressing the DIR.

Meanwhile, as evident from the clear clustering boundaries in (b) and (c) compared to Fig.1 (a) (red arrows), it is feasible to leverage the classification in helping the DIR. To fully exploit the classification in helping DIR, as we can observe from the above decomposition, the classification objective of the $p(g|x)$ can be regarded as the re-weight of the regression objective $p(y|x, g)$. Therefore, accurate estimating of the groups would be crucial as the local intra-group instance regression is also dependent on the estimated group ($g$ as the prior in $p(y|x, g)$) in our decomposed objective function. Since we divide the objective of DIR into the global group imbalance classification $p(g|x)$ and local instance imbalance regression $p(y|x, g)$, a straightforward way to solve this imbalance is to re-weight the group and instance based on their empirical label density distribution respectively.

---

[1]Groups are divided given their labels, e.g., a mapping can be formulated as $g = \lfloor \frac{y}{G} \rfloor$.

However, the data dependence (images with nearby labels) Yang et al. (2021) would hinder the accurate estimation of the real label density distribution. Motivated by Yang et al. (2021) and Parzen (1962), we investigate the semantic similarity of the DIR, which is the other aspect of the label continuity in DIR but always overlooked by previous works, and leverage the semantic similarity from both the global and local perspective to tackle both inter/intra-group imbalance and preserve the geometrical structures of the feature space.

## 3 METHODOLOGY

In this section, we propose both the symmetric and asymmetric soft labeling strategy to capture the semantic similarity and leverage the imbalance information globally across the divided groups to tackle the group imbalance. In the meantime, we introduce a label distribution smoothing to acquire the semantic similarity locally in each divided group to address the local instance imbalance.

### 3.1 SYMMETRIC DESCENDING SOFT LABELING FOR GLOBAL GROUP CLASSIFICATION

We first address the inter-group imbalance by leveraging the semantic similarity at a group level. We propose a symmetric descending soft labeling strategy to capture the semantic similarity across the groups. For a group label $g$, in the classification with cross-entropy (CE) loss, the group label is encoded into hard labels as $g_{hard} = [0, \ldots, 1_g, \ldots, 0]$ where $1_g$ denotes 1 at $g$-th index of the $g_{hard}$ list. Meanwhile, the loss function is defined as $\mathcal{L}_{CE} = -1_g \log o_g$, where $o_g$ is the output logit of the deep model at index $g$ after soft-max ($o_{pred} = \{\ldots, o_g, \ldots\}$). As we can observe from the CE, the information from the other group labels is overlooked. Consequently, only the discriminative information is learned to distinguish the groups from each other while the semantic similarity characteristics between the groups are ignored.

To tackle this problem, we propose a soft labeling strategy to capture the semantic similarity across the groups. We first define the symmetric descending soft labeling strategy to convert the group label $g$ into the soft label $g_{soft}$ :

$$l_{soft}^{sym}(g) = [\ldots, G - \beta, G, G - \beta, \ldots] \tag{1}$$

where $G$ is at the $g$-th index of the $l_{soft}^{sym}(g)$ list, $G - 1$ is at the $g \pm 1$-th index of the $l_{soft}^{sym}(g)$ list and so on, and $\beta$ is the hyper-parameter (e.g. $\beta = 1$) for distinguishing the nearby group labels. This symmetric descending soft labeling is a pyramid shaping labeling strategy with the peak at the current group label $g$ and descending symmetrically towards both two sides (index from $g$ to the start and end index).

Different from traditional soft labeling strategies Hinton et al. (2015); Díaz & Marathe (2019), our symmetric descending soft labeling strategy not only preserves the relative information between the groups, but also considers the semantic similarity from a global group perspective to deal with the group imbalance. Consequently, it is feasible for us to unleash the potential of classification for the regression task in an end-to-end manner by directly modeling $p(g|x)$.

### 3.2 ASYMMETRIC DESCENDING SOFT LABELING FOR GLOBAL GROUP CLASSIFICATION

We incorporate prior knowledge of group imbalance which is derived from the empirical group training distribution into the symmetric descending soft labeling above to tackle the imbalance classification in our objective decomposition. Instead of manually building up the groups with roughly equal numbers of data samples as Pintea et al. (2023), we count down the number of samples per group and we can obtain: $D = [D_1, D_2, \ldots, D_G]$ where $D_i$ denotes the number of samples in the group $i$. Therefore, given different levels of data imbalance across the groups, the symmetric soft labeling then becomes asymmetric.

Specifically, we calculate the inverse empirical training distribution $P_D$ from the sample count of the groups $D$ in the following way:

$$D = [D_1, D_2, \ldots, D_G] \xrightarrow{inverse} P_D = [\underbrace{1 - \frac{D_1}{\sum_i^G D_i}}_{P_1}, \underbrace{1 - \frac{D_2}{\sum_i^G D_i}}_{P_2}, \ldots, \underbrace{1 - \frac{D_G}{\sum_i^G D_i}}_{P_G}] \tag{2}$$

Then, the asymmetric descending soft labeling of a group $g$ is formulated as the following:

$$l_{soft}^{asym} = (P_D \| g) \odot l_{soft}^{sym}(g) \tag{3}$$

where $\odot$ denotes the element-wise multiplication between two vectors (the inverse empirical group training distribution and the soft labels), $P_D \| g$ denotes the symmetric soft labeling except the probability at index $g$ (the ground truth group index) in $P_D$.

For example, for a group with label $g$, the symmetric soft labeling is $l_{soft}^{symm} = [\ldots, |G|-1, |G|, |G|-1, \ldots]$, the $(P_D \| g)$ would be $(P_D \| g) = [P_1, P_2, \ldots, \hat{P}_g, \ldots, P_G]$. Instead of directly adopting the true empirical probability $P_g$ from the $P_D$, we set $\hat{P}_g = 1$ in the $(P_D \| g)$ which prevents the scaling of the current group $g$. Therefore, while the other indexes of the $l_{soft}^{sym}(g)$ list are scaled by the $P_D$ with their corresponding empirical probabilities, the index $g$ of the $l_{soft}^{asym}$ remains the same with the $l_{soft}^{sym}(g)$. As each element of the empirical probability in $P_D$ are statistically less than 1, the scaling of the symmetric soft labeling are scaling down the element not at the current ground truth index with the prior imbalance knowledge from $P_D$.

Apart from the above symmetric soft labeling, our asymmetric soft labeling not only leverages the knowledge from the whole dataset but also considers the imbalance priors of the groups. As a result, this asymmetric soft labeling can capture the semantic similarity of the groups in DIR and smooth the imbalance group distribution with the semantic similarity. By leveraging semantic similarity with our proposed soft labeling strategy, we tackle the imbalance across the groups through accurately modeling $p(g|x)$ in a end-to-end manner to unleash the potential of the classification in helping DIR.

After we have obtained the soft labels, we then forward the soft labels into the soft-max. The final prediction logits of the $l_{soft}^{asym}$ after soft-max would become soft-max $(l_{soft}^{asym}) = [q_1, \ldots, q_G]$ with $\sum_i^G q_i = 1$. Therefore, the classification loss (NLL of $p(g|x)$ for an instance is formulated as:

$$\mathcal{L}_{cls} = -\sum_{i=1}^{G} q_i \log o_i \tag{4}$$

noting that this loss is calculated on every index of the logits from index 1 to $G$.

**Understanding why our soft labeling strategy can help to address DIR.** By proposing the both symmetric and asymmetric soft labeling strategy to the DIR, we bridge the gap from the $p(z|x)$ to $p(g|)$, which is an end-to-end solution to address the imbalance across the groups in the objective of DIR. Also, our soft labeling strategy can be regarded as a global knowledge smoothing for the groups. As stated in Chen et al. (2021), the divergences of the feature norm between different training distributions are the main reason that hinders the adaptation from the imbalanced training to the balanced testing. However, Müller et al. (2020) observed that label smoothing can effectively reduce the feature norms. Based on this observation, our proposed global group label smoothing can constraint on the feature norms of data instance in the groups and preserve the geometrical structures of the feature space by leveraging the knowledge from similar data, which can better handle the distribution divergence between the training and testing distributions and help to address the imbalance across the groups. Therefore, by leveraging the soft labeling strategy to model the $p(g|x)$, we can unleash the power of classification in helping DIR.

### 3.3 LABEL DISTRIBUTION SMOOTHING FOR LOCAL INSTANCE

In order to capture the semantic similarity for the local intra-group instance, inspired by Yang et al. (2021), we introduce the label distribution smoothing (LDS) for each group of data. Specifically, in LDS, a symmetric kernel (e.g. Gaussian kernel) is adopted to borrow the feature at nearby labels to redeem for the data imbalance. The smoothed label density of one arbitrary instance $(x, y, g)$ can be written as the following:

$$\hat{p}(y) = \int_{y' \in Y} k(y', y) p(y') dy' \tag{5}$$

Then, the mean-square error (MSE) loss of this instance can be written as follows:

$$\mathcal{L}_{MSE}^g = \hat{p}(y)(y - y_{pred})^2 \tag{6}$$

where the MSE loss is calculated on the head $g$ of the multi-head regressor. The total MSE over all $G$ groups is formulated as:

$$\mathcal{L}_{MSE} = \sum_{g=1}^{G} \mathcal{L}_{MSE}^{g} \tag{7}$$

As we can observe from Equation 5, LDS incorporates the semantic similarity of the data instance at nearby labels. Compared to soft labeling which leverages global semantic similarity across the entire dataset, LDS focuses on local semantic similarity among neighboring labels. Therefore, our proposed method can tackle the DIR in a coarse to fine-grained, global-to-local manner.

### 3.4 GROUP CONTRASTIVE REPRESENTATION LEARNING

In order to fully exploit the potential of the classification, we take advantage of the representation learning to learn an imbalance-robust feature representation to build up a solid foundation for both group classification and local instance regression. Considering the fact that our downstream tasks of DIR (modeling $p(g|x)$ and $p(y|x,g)$) both involve the group classification and group-guided multi-heads regression, learning a group-level imbalance-robust feature is crucial for our downstream tasks. Inspired by Zha et al. (2023) and in order to further leverage the classification for the DIR, we perform the contrastive learning with respect to the groups and formulate the group contrastive loss (GCL) as the following:

$$\mathcal{L}_{GCL} = -\frac{1}{B(B-1)} \sum_{i=1}^{B} \sum_{\substack{j=1, \\ j \neq i}}^{B} \log \frac{s(z_i, z_j)}{\sum_{k=1}^{B} \mathbf{1}_{[k \neq i, d(g_i, g_k) \geq d(g_i, g_j)]} s(z_i, z_k)} \tag{8}$$

where for the index $i, j, k$ of three arbitrary instance index in the batch, $s(i, j)$ denotes the abbreviate of $exp(sim(z_i, z_j)/t)$, $sim(\cdot)$ denotes the similarity function, $d(\cdot)$ denotes the distance function, and $exp(\cdot)$ is the exponential function. Following Zha et al. (2023), we use cosine similarity as the $sim(\cdot)$ and L1 distance as the $d(\cdot)$. Moreover, $\mathbf{1}$ denotes the zero-one indicator, $t$ denotes the temperature hyper-parameter, and $B$ is the batch size.

### 3.5 CLASSIFICATION-GUIDED MULTI-HEADS REGRESSION

We formulate the training and inference procedures in this section to show how can we leverage the classification to help DIR from end to end. During training, we first train the feature encoder based on the Equation 8. Then, the feature representations would be fed into the classification head to make the estimation of which group the feature representation should be by penalizing with the Loss 4 with the Soft Labels 3. Simultaneously, given the ground truth group label, the feature representations would be forwarded to their corresponding regressor heads with the MSE loss as 6. At the inference phase, after the feature extraction, we first predict the group labels from the classification head and then obtain the results at the regressor heads from the previous prediction.

## 4 EXPERIMENTS

### 4.1 IMPLEMENTATION DETAILS

We implement our proposed method on three real-world benchmarks, AgeDB-DIR, IMDB-WIKI-DIR, and STS-B-DIR. To make a fair comparison, as Yang et al. (2021); Zha et al. (2023); Zhang et al. (2023a) we used ResNet-18 as the backbone for AgeDB-DIR, ResNet-50 as the backbone for IMDB-WIKI-DIR. For the STS-B-DIR, we used BiLSTM + 300 D (dimension) GloVe word embedding as the backbone and the word processing tool to embed each word into a 300-dimension vector. For the classification head, we adopted a linear layer with $G$ output neurons to make the $G$-class classification. For the multi-heads regressor, we used a linear layer of the $G$ output neurons where each output neuron is corresponding to an independent regressor. We use the mean absolute error (MAE) and geometric mean (GM) as the measurement of the performance of our proposed method for AgeDB-DIR and IMDB-WIKI-DIR dataset. Mean square error (MSE) and Pearson Correlation for the STS-B-DIR dataset. Specifically, we count down the number of instances into different shots

(majority $> 100$ /median $20 \sim 100$/few shots $< 20$) and calculate the above measurements over each shot to make a more comprehensive analysis.

## 4.2 REAL-WORLD DATASETS

We validate the effectiveness of our proposed method based on the three real-world benchmarks which has been curated by Yang et al. (2021) for the DIR task.

**AgeDB-DIR** Moschoglou et al. (2017) is a human facial dataset that contains 12.2K training images, 2.1K testing images and 2.1K validation images. The label of the image is their corresponding age. The minimum of age is 0 and the maximum is 101.

**IMDB-WIKI-DIR** Rothe et al. (2016) is also another large facial datasets collected from the Internet (IMDB-WIKI). It contains 191.5 K training samples, 11K testing samples and 11K validation samples. The number of samples per label varies from 0 to 7149. The minimum of the age is 1 and the maximum is 186. The task is to estimate the age from the input images.

**STS-B-DIR** Cer et al. (2017) is a semantic textual similarity benchmark which measures the similarity between any arbitrary two-sentence pair collected from video, news headlines and so on. It contains 5.2K training pairs, 1K testing pairs and 1K validation pairs. The measures vary from 1 to 5 and the granularity is 0.1 for each label. The task is to estimate the similarity of each pair.

## 4.3 ANALYSIS OF AgeDB-DIR

As we can observe from Table 1, our proposed method symmetric soft labeling strategy can outperform other methods in overall MAE (0.05 better than Zha et al. (2023), 0.19 better than Wang & Wang (2023), and at least 0.2 better than other DIR solutions). Specifically, we have a 1.1 improvement compared to the vanilla, 0.9 improvements over the Yang et al. (2021), 0.8 improvements over the Zhang et al. (2023a) and 0.4 improvements over the Gong et al. (2022) In the meantime, our proposed asymmetric soft labeling strategy which leveraged the imbalance information from the training distribution significantly outperforms the symmetric soft labeling strategy. Compared to other DIR solutions, the overall MAE has a at least 0.18 improvement and the majority MAE has a at least 0.36 improvement. Additionally, the GM of the majority in asymmetric soft labeling strategy also outperform the others, which shows the a better prediction fairness in the majority shot and consequently exhibits a better performance in the overall MAE.

Moreover, compared to Fig.1 (b) and (a), our proposed symmetric and asymmetric soft labeling strategy can better maintain the geometrical structure of the feature space as shown in Fig.2 (a) and (b) compared to Fig.2 (c). The asymmetric soft labeling strategy (Fig.2 (b)) would induce the feature space not only ordinal as the (a) in Fig.1, but also shows a more obvious cluster boundary than the symmetric soft labeling strategy (Fig.2 (b)). As we can observe from Fig.2, the symmetric and asymmetric soft labeling strategy can better capture the geometric structure than the fine-tuning in Fig.1(b) and (c). Furthermore, it showcases that the asymmetric soft labeling strategy can better leverage the classification compared to the symmetric soft labeling strategy and better capture the geometric structure than the direct fine tuning (Fig.2 (c)). This is because our soft labeling strategy can be regraded as a smoothing strategy that leverages the information from the nearby labels (both group and instance), which can unleash the potential of classification in helping the DIR.

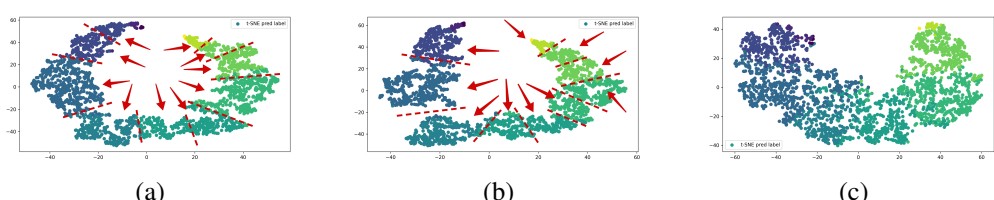

|     (a)     |     (b)     |     (c)     |

Figure 2: The t-SNE (AgeDB-DIR, 10 groups demo) of the features under (a) symmetric soft labeling (b) asymmetric soft labeling (c) cross-entropy (CE).

Table 1: Evaluation on AgeDB-DIR.

| Shot / Method | MAE↓ | | | | GM↓ | | | |
|---|---|---|---|---|---|---|---|---|
| | All | Many. | Med. | Few. | All | Many. | Med. | Few. |
| VANILLA | 7.77 | 6.62 | 9.55 | 13.67 | 5.05 | 4.23 | 7.01 | 10.75 |
| SMOTER Torgo et al. (2013) | 8.16 | 7.39 | 8.65 | 12.28 | 5.21 | 4.65 | 5.69 | 8.49 |
| SMOGN Branco et al. (2017) | 8.26 | 7.64 | 9.01 | 12.09 | 5.36 | 4.90 | 6.19 | 8.44 |
| RRT Kang et al. (2020) | 7.74 | 6.98 | 8.79 | 11.99 | 5.00 | 4.50 | 5.88 | 8.63 |
| RRT+LDS Yang et al. (2021) | 7.72 | 7.00 | 8.75 | 11.62 | 4.98 | 4.54 | 5.71 | 8.27 |
| FOCAL-R Lin et al. (2017) | 7.64 | 6.68 | 9.22 | 13.00 | 4.90 | 4.26 | 6.39 | 9.52 |
| SQINV Yang et al. (2021) | 7.81 | 7.16 | 8.80 | 11.20 | 4.99 | 4.57 | 5.73 | 7.77 |
| SQINV + LDS Yang et al. (2021) | 7.67 | 6.98 | 8.86 | 10.89 | 4.85 | 4.39 | 5.80 | 7.45 |
| LDS+FDS Yang et al. (2021) | 7.55 | 7.01 | 8.24 | 10.79 | 4.72 | 4.36 | 5.45 | 6.79 |
| LDS+FDS+DER Amini et al. (2020) | 8.18 | 7.44 | 9.52 | 11.45 | 5.30 | 4.75 | 6.74 | 7.68 |
| VAE Kingma & Welling (2013) | 7.63 | 6.58 | 9.21 | 13.45 | 4.86 | 4.11 | 6.61 | 10.24 |
| RANKSIM Gong et al. (2022) | 7.02 | 6.49 | 7.84 | 9.68 | 4.53 | 4.13 | 5.37 | 6.89 |
| OE Zhang et al. (2023a) | 7.46 | 6.73 | 8.18 | 12.38 | 4.72 | 4.21 | 5.36 | 9.70 |
| Con-R Keramati et al. (2024) | 7.20 | 6.50 | 8.04 | 9.73 | 4.59 | 3.94 | **4.83** | 6.39 |
| VIR Wang & Wang (2023) | 6.99 | 6.39 | 7.47 | **9.51** | 4.41 | 4.07 | 5.05 | **6.23** |
| SupCR Zha et al. (2023) | 6.85 | 6.20 | 7.62 | 10.82 | 4.32 | 3.89 | 4.95 | 8.02 |
| Ours (Symmetric) | 6.81 | 6.18 | **7.44** | 10.27 | **4.30** | 3.81 | 5.27 | 6.55 |
| Ours (Asymmetric) | **6.67** | **5.84** | 7.96 | 10.85 | 4.37 | **3.67** | 5.79 | 7.73 |

Table 2: Evaluation on IMDB-WIKI-DIR.

| Shot / Method | MAE↓ | | | | GM↓ | | | |
|---|---|---|---|---|---|---|---|---|
| | All | Many. | Med. | Few. | All | Many. | Med. | Few. |
| VANILLA | 8.06 | 7.23 | 15.12 | 26.33 | 4.57 | 4.17 | 10.59 | 20.46 |
| SMOTER Torgo et al. (2013) | 8.14 | 7.42 | 14.15 | 25.28 | 4.64 | 4.30 | 9.05 | 19.46 |
| SMOGN Branco et al. (2017) | 8.03 | 7.30 | 14.02 | 25.93 | 4.63 | 4.30 | 8.74 | 20.12 |
| SMOGN + LDS Yang et al. (2021) | 8.02 | 7.39 | 13.71 | 23.22 | 4.63 | 4.39 | 8.71 | 15.80 |
| RRT Kang et al. (2020) | 7.81 | 7.07 | 14.06 | 25.13 | 4.35 | 4.03 | 8.91 | 16.96 |
| RRT+LDS Yang et al. (2021) | 7.79 | 7.08 | 13.76 | 24.64 | 4.34 | 4.02 | 8.72 | 16.92 |
| SQINV+LDS Yang et al. (2021) | 7.83 | 7.31 | 12.43 | 22.51 | 4.42 | 4.19 | 7.00 | 13.94 |
| FOCAL-R Lin et al. (2017) | 7.97 | 7.12 | 15.14 | 26.96 | 4.49 | 4.10 | 10.37 | 21.20 |
| FOCAL-R+LDS Yang et al. (2021) | 7.90 | 7.10 | 14.72 | 25.84 | 4.47 | 4.09 | 10.11 | 19.14 |
| BMCRen et al. (2022) | 8.08 | 7.52 | 12.47 | 23.29 | - | - | - | - |
| GAIRen et al. (2022) | 8.12 | 7.58 | 12.27 | 23.05 | - | - | - | - |
| VAE Kingma & Welling (2013) | 8.04 | 7.20 | 15.05 | 26.30 | 4.57 | 4.22 | 10.56 | 20.72 |
| RANKSIM Gong et al. (2022) | 7.50 | 6.93 | 12.09 | 21.68 | 4.19 | 3.97 | 6.65 | 13.28 |
| DER Amini et al. (2020) | 7.85 | 7.18 | 13.35 | 24.12 | 4.47 | 4.18 | 8.18 | 15.18 |
| LDS + FDS + DER Amini et al. (2020) | 7.24 | 6.64 | 11.87 | 23.44 | 3.93 | 3.69 | 6.64 | 16.00 |
| Con-R Keramati et al. (2024) | 7.33 | 6.75 | 11.99 | 22.22 | 4.02 | 3.79 | 6.98 | 12.95 |
| VIR Wang & Wang (2023) | 7.19 | 6.56 | 11.81 | 20.96 | **3.85** | **3.63** | 6.51 | 12.23 |
| Ours (Symmetric) | 7.22 | 6.70 | **10.72** | **20.35** | 3.87 | 3.68 | **5.74** | **11.14** |
| Ours (Asymmetric) | **7.18** | **6.55** | 11.42 | 20.87 | 3.91 | 3.66 | 6.69 | 13.07 |

## 4.4 ANALYSIS OF IMDB-WIKI-DIR

As we can observe from Table 2, our proposed method can perform better than other DIR solutions in overall MAE. Compared to LDS and FDS, our method has a ∼0.8 improvement on MAE. Compared to Balanced-MSE, our method has a ∼0.8 improvement on MAE. Also, we have a 0.32 improvement on RANKSIM and a 0.15 improvement on MAE. As for the symmetric soft labeling strategy, the median and few shots are always better than other methods in MAE. When we compare the GM to other methods, the symmetric soft labeling strategy also performs better than others on the median and few shots. For the asymmetric soft labeling strategy, the majority shot always outperforms than other methods in MAE. In summary, our proposed method can perform better than most of the solutions in DIR. As a results, this also showcases that our soft labeling strategy can capture the semantic similarity to unleash the potential of the classification in helping DIR in all majority, median and few shots.

## 4.5 ANALYSIS OF STS-B-DIR

We show the performance of our proposed method on STS-B-DIR in Table.3. As we can observe from Table.3, our proposed method can also achieve a state-of-art performance in both symmetric

Table 3: Evaluation on STS-B-DIR.

| Shot | MSE↓ | | | | Pearson Correlation↑ | | | |
|---|---|---|---|---|---|---|---|---|
| Method | All | Many. | Med. | Few. | All | Many. | Med. | Few. |
| VANILLA | 0.974 | 0.851 | 1.520 | 0.984 | 74.2 | 72.0 | 62.7 | 75.2 |
| SMOTER Torgo et al. (2013) | 1.046 | 0.924 | 1.542 | 1.154 | 72.6 | 69.3 | 65.3 | 70.6 |
| SMOGN Branco et al. (2017) | 0.990 | 0.896 | 1.327 | 1.175 | 73.2 | 70.4 | 65.5 | 69.2 |
| SMOGN + LDS Yang et al. (2021) | 0.962 | 0.880 | 1.242 | 1.155 | 74.0 | 71.5 | 65.2 | 69.8 |
| RRT Kang et al. (2020) | 0.964 | 0.842 | 1.503 | 0.978 | 74.5 | 72.4 | 62.3 | 75.4 |
| FOCAL-R Lin et al. (2017) | 0.951 | 0.843 | 1.425 | 0.957 | 74.6 | 72.3 | 61.8 | 76.4 |
| INV Yang et al. (2021) | 1.005 | 0.894 | 1.482 | 1.046 | 72.8 | 70.3 | 62.5 | 73.2 |
| INV + LDS Yang et al. (2021) | 0.914 | 0.819 | 1.31 | 0.95 | 75.6 | 73.4 | 63.8 | 76.2 |
| VAE Kingma & Welling (2013) | 0.968 | 0.833 | 1.511 | 1.102 | 75.1 | 72.4 | 62.1 | 74.0 |
| DER Amini et al. (2020) | 1.001 | 0.912 | 1.368 | 1.055 | 73.2 | 71.1 | 64.6 | 74.0 |
| LDS Yang et al. (2021) | 0.914 | 0.819 | 1.319 | 0.955 | 75.6 | 73.4 | 63.8 | 76.0 |
| FDS Yang et al. (2021) | 0.927 | 0.851 | 1.225 | 1.012 | 75.0 | 72.4 | 66.7 | 74.2 |
| VIR Wang & Wang (2023) | 0.892 | **0.795** | 0.899 | 0.781 | 77.6 | 75.2 | 69.6 | **84.5** |
| LDS + FDS Yang et al. (2021) | 0.907 | 0.802 | 1.363 | 0.942 | 76.0 | 74.0 | 65.2 | 76.6 |
| RANKSIM Gong et al. (2022) | 0.903 | 0.908 | 0.911 | 0.804 | 75.8 | 70.6 | 69.0 | 82.7 |
| LDS + FDS + DER Amini et al. (2020) | 1.007 | 0.880 | 1.535 | 1.086 | 72.9 | 71.4 | 63.5 | 73.1 |
| Ours (Symmetric) | **0.885** | 0.801 | **0.887** | **0.779** | **77.8** | 75.3 | **69.9** | 84.1 |
| Ours (Asymmetric) | 0.893 | 0.799 | 0.894 | 0.782 | 77.5 | **75.4** | 67.7 | 82.9 |

and asymmetric soft labeling strategies. Compared to Yang et al. (2021), our symmetric and asymmetric strategy can have a ∼0.1 improvement on the MSE and ∼1.8% improvement on the Pearson correlation. Compared to Gong et al. (2022), our symmetric and asymmetric strategy can also have a ∼0.015 improvement on the MSE and ∼2% improvement on the Pearson correlation. Compared to Wang & Wang (2023), our symmetric strategy can have a ∼0.001 improvement on the MSE and ∼0.2% improvement on the Pearson correlation. Interestingly, in STS-B-DIR, the symmetric soft labeling strategy outperforms asymmetric strategy in the overall, this is because the imbalance of the STS-B-DIR is not as severe as the AgeDB-DIR and IMDB-WIKI-DIR and the number of instance in majority shots is close to the median shots and the few shots.

### 4.6 ABLATION STUDY

To further explain the effectiveness of our proposed method, we conduct the ablation study on different numbers of groups in Fig.3. Compared to the CE loss, our proposed method can achieve a better performance over the group classification. As we stated in our methodology, CE can provide no other information when calculating the group classification loss and ignore the semantic similarity across the groups. Therefore, as we can observe from Fig.3(a), the classification performance of CE would always be worse than the symmetric and asymmetric strategy. Moreover, when we observe Fig.3(b), the MAE of CE is also a lot worse than the symmetric and asymmetric strategy. In Fig.3(c), the G-Mean performance of CE is also worse than the symmetric and asymmetric soft labeling. showcasing that semantic similarity is a crucial aspect of data continuity in DIR.

Furthermore, when we compare the group numbers across these three classification losses, as we can observe from Fig.3(a), the group classification accuracy drops. This is because with the increasing of the group numbers, the nearby groups would be more similar, which makes the classifier harder and harder to distinguish. However, when we leverage the semantic similarity for the group classification, our proposed solution would smooth the discrepancy between the groups. Consequently, our method can outperform the CE and perform steadily over the different number of groups (Fig.3(b)), which further validates the effectiveness of our proposed method.

## 5 RELATED WORK

### 5.1 IMBALANCED CLASSIFICATION

Imbalanced classification is a widely explored problem in the field of machine learning Zhang et al. (2023b). The solution of imbalanced classification can be concluded as the following perspectives. Firstly, re-weighting Cui et al. (2019); Jamal et al. (2020); Chu et al. (2020); He & Garcia (2009); Kim et al. (2020); Huang et al. (2016); Branco et al. (2017) is the most popular solution for the imbalanced classification. Secondly, post-hoc methods Ren et al. (2020); Tian et al. (2020); Menon

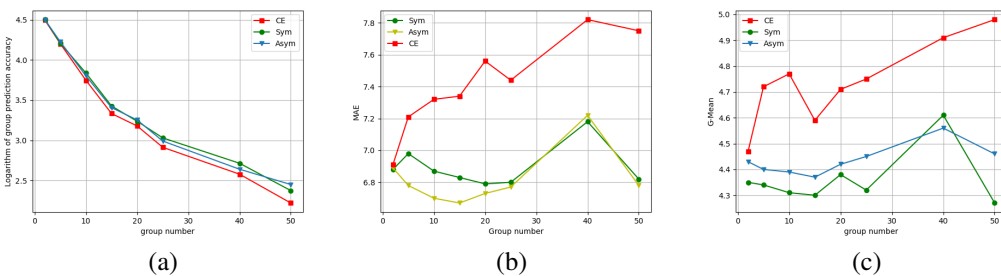

(a)                  (b)                  (c)

Figure 3: The ablation study on AgeDB-DIR, 10 groups demo of different number of groups (a) group prediction accuracy (b) MAE (c) G-Mean (GM).

et al. (2021) which aim to calibrate the predicted logits with the class prior has been shown to be an effective solution for addressing the imbalance in classification. Thirdly, mixture-of-experts Zhou et al. (2022), feature selections Han et al. (2022) and instance difficulty measuring Yu et al. (2022) are also effective solutions for imbalanced classification. Moreover, representation learning Liu et al. (2019); Dong et al. (2017); Wang et al. (2021); Li et al. (2022); Liu et al. (2021) with data augmentation Liu et al. (2020); Yang et al. (2022); Chou et al. (2020b); Huang et al. (2016); Shi et al. (2022) (e.g. MixUp Chou et al. (2020a)) is also a feasible solution for the imbalanced classification by learning imbalance-robust feature representations.

### 5.2 DEEP IMBALANCED REGRESSION

Deep imbalanced regression (DIR) has been proposed by Yang et al. (2021) and has attracted tremendous interests in the recent machine learning studies. Similarly, re-weighting Torgo et al. (2013); Steininger et al. (2021); Branco et al. (2018); Stocksieker et al. (2023) has also been adopted in the regression tasks. Yang et al. (2021) proposed a label distribution smoothing and feature distribution smoothing to redeem the imbalance. Ren et al. (2022); Silva et al. (2022) revised the MSE loss for accommodating the imbalance distribution. Jiang et al. (2023) used a mixture-of-experts on the outputs while Wu et al. (2023); Yao et al. (2022) proposed a mix-up strategy for dealing with the regression tasks. Wang & Wang (2023) used the variational inference for addressing the DIR. Moreover, integrating classification regularizers with the mean squre error loss (MSE) Gong et al. (2022); Zha et al. (2023); Zhang et al. (2023b;a); Keramati et al. (2024); Pintea et al. (2023) has been empirically shown to be effective in tackling the DIR. Different from previous works, instead of directly incorporating the classification regularizers, our work aims to unleash the power of the classification for better helping the DIR by capturing the semantic similarity.

## 6 CONCLUSION

In this paper, we investigate the semantic similarity, a characteristic which has been always overlooked in previous works, to unleash the potential of the classification in helping DIR. Specifically, we decompose the imbalance of DIR into global and local imbalances. We propose a symmetric and asymmetric soft labeling strategy that captures the semantic similarity to tackle the global group imbalance. Furthermore, we use the label distribution smoothing to handle the local instance imbalance. By linking up the global group classification with the local instance regression, we unleash the potential of the classification and solve the DIR from end-to-end. Extensive experiments over real-world datasets also validate the effectiveness of our proposed method.

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
