# Unleashing the Potential of Classification with Semantic Similarity for Deep Imbalanced Regression

## 1 Supplementary

### 1.1 Additional Implementation Details

To make a fair comparison, the vanilla model we used in our experiments for the Agedb-DIR is ResNet-18 and ResNet-50 for IMDB-WIKI-DIR. Also, we take ResNet-18 and 50 as our backbone while we revised the output dimension from 1-d (single output) to a $G-d$ classification head and a multi-heads ($G$-heads) regressor for our method. Also, we use Mean square error (MSE) and Mean absolute error (MAE) as the loss function for AgeDB-DIR, IMDB-WIKI-DIR, and STS-B-DIR for the fine tuning. Adam is employed as the optimizer, the momentum is set as 0.9 and the weight decay is 1e-4. We used a grid search for the learning rate which varies from $\{1e^{-2}, 1e^{-3}, 5e^{-5}, 1e^{-4}, 5e^{-4}\}$ for AgeDB-DIR and STS-B-DIR, $\{1e^{-4}, 1e^{-5}, 1e^{-6}, 1e^{-7}, 1e^{-8}\}$ for IMDB-WIKI-DIR.

Furthermore, we set the temperature parameter to 2.5 by conducting a grid search from $1-5$. We recommend $\geq 1$ temperature parameter for the increasing of batch size (the batch size is set to 128 in our experiment due to the GPU limitations). The contrastive encoder was trained for 300 epochs, which is 100 epochs less than Zha et al. (2023) as we aim to train a group-level contrastive learning at a group-level granularity (instead of an instance level as Zha et al. (2023)), the classification & regression head are trained in a grid search manner from $\{50, 60, 70, 80, 90\}$, and the group number is also from $\{2, 5, 10, 15, 20, 25, 40, 50\}$. During training the encoder, the classification head and regression head are fixed, and we use the $\mathcal{L}_{MSE}$ and $\mathcal{L}_{cls}$ as the loss in the fine-tuning phase.

### 1.2 Ablation Study and Analysis on Different Classification Criterion

In order to validate the necessity of leveraging the semantic similarity in helping DIR, we conduct experiments on the AgeDB-DIR with several imbalanced classification solutions in Table.1. We can observe from Table.1, that the imbalanced classification solutions are undermined in DIR. This is because that the data dependence in DIR Yang et al. (2021) would hinder the accurate estimation of the groups. These imbalanced classification solutions would ignore the information from the other classes while only focusing on the current class. As a result, other information from other classes would not be learned and the semantic similarity across the groups are overlooked.

As we can observe from Table.1, the soft labeling strategy can outperform the imbalanced classification solutions a lot. This shows the effectiveness of our proposed method. Specifically, the soft labeling strategy can leverage the information from the nearby groups to redeem the group imbalance and take the imbalance priors into consideration. Therefore, our proposed method is uniquely designed and especially effective for the DIR compared to the previous imbalanced classification solutions. We also show the test error (L1) loss on each label between the vanilla model and our proposed method (noting that the y-axis is different between two images.) in Fig.2 and the number of training samples each label in Fig.1 to demonstrate the effectiveness of our proposed method.

## References

Cong et al. Cong. Decoupled optimisation for long-tailed visual recognition. *Proceedings of the AAAI Conference on Artificial Intelligence*, 2024.

Table 1: Comparison between different SOTA imbalance group (#) classification (%) solutions for AgeDB-DIR.

| # | Soft labeling | LAMenon et al. (2021) | NC Yang (2022) | DO Cong (2024) | ReBAT Wang (2024) |
|---|---|---|---|---|---|
| 5 | 67.71 | 55.71 | 66.01 | 66.72 | 67.04 |
| 10 | 39.99 | 29.53 | 38.75 | 38.41 | 39.29 |
| 25 | 19.53 | 16.68 | 18.43 | 19.07 | 19.14 |
| 50 | 10.09 | 8.83 | 9.67 | 9.44 | 9.78 |

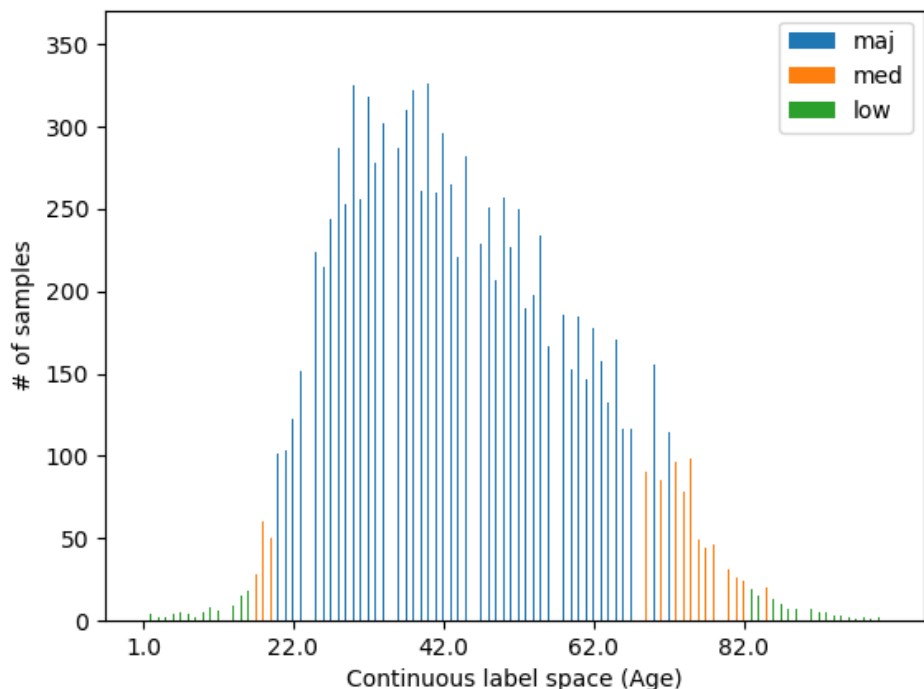

Figure 1: The ground truth for the number of samples in training with respect to different shots.

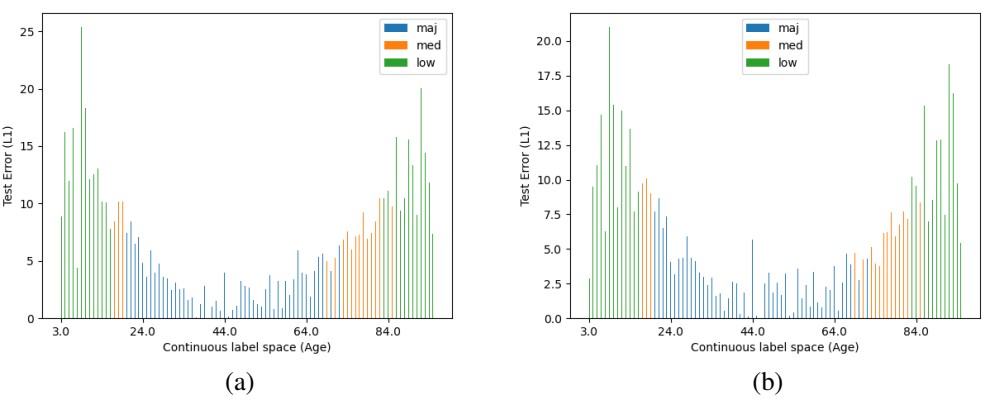

Figure 2: The test error (L1) in (a) vanilla regression model, (b) soft labeling strategy based regression model. Different color denotes the different shots (majority/median/few shots). (Noting that the y-axis is different in two images)

Aditya Krishna Menon, Sadeep Jayasumana, Ankit Singh Rawat, Himanshu Jain, Andreas Veit, and Sanjiv Kumar. Long-tail learning via logit adjustment. In *International Conference on Learning Representations*, 2021. URL `https://openreview.net/forum?id=37nvvqkCo5`.

Yifei et al. Wang. Balance, imbalance, and rebalance: Understanding robust overfitting from a minimax game perspective. *Advances in neural information processing systems*, 2024.

Yibo et al. Yang. Inducing neural collapse in imbalanced learning: Do we really need a learnable classifier at the end of deep neural network? *Advances in neural information processing systems*, 2022.

Yuzhe Yang, Kaiwen Zha, Yingcong Chen, Hao Wang, and Dina Katabi. Delving into deep imbalanced regression. In *International conference on machine learning*, pp. 11842–11851. PMLR, 2021.

Kaiwen Zha, Peng Cao, Jeany Son, Yuzhe Yang, and Dina Katabi. Rank-n-contrast: Learning continuous representations for regression. In *Thirty-seventh Conference on Neural Information Processing Systems*, 2023.