# OpenReview forum: "Unleashing the Potential of Classification with Semantic Similarity for Deep Imbalanced Regression"
_ICLR.cc/2025/Conference — Submitted to ICLR 2025_

### Official Review · Reviewer_bTbt · 2024-10-30

**Soundness:** 2
**Presentation:** 2
**Contribution:** 2
**Rating:** 5
**Confidence:** 4

**Summary:**

The authors of this paper concern with the label imbalance issue in regression tasks. They have observed that directly incorporating classification regularizers widely used in current methods may potentially distort the geometrical structure of the feature space due to the label imbalance. To address this issue, they propose both symmetric and asymmetric soft labeling strategies to capture semantic similarity so as to handle the label imbalance. The experimental results empirically show the effectiveness of the proposed soft labeling strategy.

**Strengths:**

-	The label imbalance issue is important for regression tasks and the phenomenon of distorting the geometrical structure of the feature space due to label imbalance observed in this paper seems to be interesting.
-	The experimental results and ablation study show the effectiveness of the proposed soft labeling strategy.

**Weaknesses:**

-	*The motivation is unclear.* First, the description of deep imbalance regression is too simple and it is difficult for some readers to understand the task and its scenarios. Some visual examples are expected in the introduction. Second, the phenomenon of distorting the geometrical structure of the feature space seems to be interesting, but how and why does it affect the performance of deep imbalance regression? Finally, while the authors present two soft labeling strategies based on semantic similarity to smooth the labels, they do not directly address the label imbalance. It would be better to clarify how the semantic similarity and the proposed method specifically deal with the label imbalance. Otherwise, the proposed label smoothing based on semantic similarity modifies the original one-hot label vector to a soft label. It may introduce some noises, so how do the authors deal with those noisy supervision?
-	*The contribution may be incremental.* The paper only proposes two soft labeling strategies for group classification from the label smoothing perspective, and employs existing label distribution smoothing [1] and group contrastive representation learning [2] tricks. So, the contribution seems to be incremental.
-	*Some theoretical analyses are required.* The paper could be strengthened by providing some theoretical analysis and insights to support the empirical findings.
-	*Minor issues.* (1) The definition of $c(\psi)$ is missing. (2) “accurate estimating” in line 156 should be “accurate estimation”. (3) The figures in this paper are unclear.

[1] Yuzhe Yang, Kaiwen Zha, Yingcong Chen, Hao Wang, and Dina Katabi. Delving into deep imbalanced regression. In International conference on machine learning, pp. 11842–11851. PMLR, 2021.

[2] Kaiwen Zha, Peng Cao, Jeany Son, Yuzhe Yang, and Dina Katabi. Rank-n-contrast: Learning
continuous representations for regression. In Thirty-seventh Conference on Neural Information
Processing Systems, 2023.

**Questions:**

-	From section 3.5, the training procedure consists of two stages, i.e., training feature encoder and inducing multiple regressor heads, which may result in a suboptimal model. Why not train feature encoder and multiple regressor heads jointly as [1]?
-	What is the experimental setting of cross-entropy?

[1] Mahsa Keramati, Lili Meng, and R. David Evans. Conr: Contrastive regularizer for deep imbalanced regression. In The Twelfth International Conference on Learning Representations, 2024.

---

### Official Review · Reviewer_Fz1p · 2024-10-30

**Soundness:** 2
**Presentation:** 3
**Contribution:** 2
**Rating:** 5
**Confidence:** 3

**Summary:**

This paper highlights that previous works primarily focus on learning the ordinal characteristics of data within the feature space, but overlook that the similarity across labels can reflect the similarity of the corresponding features. The authors propose a framework that decomposes the regression task into group classification and instance regression within each group. They customize symmetric group labels and derive asymmetric group labels based on the statistics of the samples within the group. A group classifier predicts the sample group labels and calculates the classification loss using these asymmetric group labels. For samples within the group, inspired by Yang et al. (2021), the authors employ smoothed labels to calculate the regression loss.

**Strengths:**

1. Clear problem definition.
2. Adequate introduction to related work.
3. Logical and concise writing.

**Weaknesses:**

I have several concerns in the following aspects:
1. Previous work has employed smoothed labels to calculate the mean squared error (MSE) for training regression models. The authors introduce a group classifier based on this concept; however, according to the article, the group classifier operates independently from the instance regressor that predicts \( y \). The group label is not directly related to the sample label, and the authors have not provided the generation process or conducted an ablation experiment for the group label. More experiments and inferences are needed to demonstrate the necessity of the proposed group classifier.

2. Some writing needs better explanation:
- The legend in Fig. 1 requires clearer color interpretation.
- The definition of the sign \( k \) in Eq. 5 is missing.
- The final training objective needs clarification.
- The presence of peaks in Fig. 3(b) and (c) when the group number is 40 needs an explanation.
- Line 247: Missing part of \( p(g|) \).
- Line 324: Lack of space after the '/'.

3. Further ablation studies are recommended:
- An ablation study excluding the classification term is needed.
- An ablation study excluding contrastive learning should also be included.

**Questions:**

1. A group contains several classes. On what basis are the same group labels assigned to each class within the group?
2. How can you explain the statement "our asymmetric soft labeling not only leverages the knowledge from the whole dataset"? According to Eq. 3, only the group statistics are included in the calculation.
3. As mentioned in line 249, the normalized features may hinder adaptation. What is the performance of the features without normalization?
4. How does the semantic similarity \( \hat{p}(y) \) assist in addressing imbalanced regression? In previous work, "Daso: Distribution-aware semantics-oriented pseudo-label for imbalanced semi-supervised learning," a semantic classifier is biased towards the tail classes. Is there a relationship between this work and the current study?

---

### Official Review · Reviewer_Kyvn · 2024-10-31

**Soundness:** 2
**Presentation:** 2
**Contribution:** 2
**Rating:** 5
**Confidence:** 4

**Summary:**

This paper explores semantic similarity to enhance the effectiveness of classification in DIR. Specifically, the imbalance in DIR is decomposed into global and local imbalances, and a symmetric and asymmetric soft labeling strategy is proposed. This strategy captures semantic similarity to address the global group imbalance effectively.

**Strengths:**

1. An end-to-end approach decomposes the DIR objective into a global between-group imbalance classification task and a local within-group imbalance regression task. Semantic similarity is leveraged through symmetric and asymmetric soft labeling strategies and label distribution smoothing, marking an innovative shift from traditional methods that embed classification regularizers directly in feature space.
2. The paper is well-structured and logically clear, with accessible methodology, making it easier for readers to follow.
3. The authors have open-sourced their code, demonstrating a commendable commitment to research transparency and accessibility.

**Weaknesses:**

1. The methodology includes pseudo-labeling, label smoothing, and contrastive learning, which are widely used techniques. While the symmetric descending soft labeling strategy is interesting, it may not sufficiently support the novelty of the entire study.
2. Soft labeling plays a crucial role in the algorithm. How does the paper ensure the generation of high-quality soft labels?
3. There are differences in the comparison algorithms across the three datasets, with certain advanced baselines missing from the IMDB-WIKI-DIR and STS-B-DIR datasets.
4. Details on some hyperparameter settings are lacking. The paper does not specify values for $\beta$ in the soft label settings and the batch size for contrastive learning. Additionally, the temperature coefficient for contrastive learning is not explicitly shown in the equation.

**Questions:**

See the above weaknesses.

---

### Official Review · Reviewer_dier · 2024-11-01

**Soundness:** 1
**Presentation:** 2
**Contribution:** 1
**Rating:** 3
**Confidence:** 5

**Summary:**

This paper proposes splitting Deep Imbalanced Regression into global group classification and local instance regression tasks. Soft labelling and label distribution smoothing are used to address imbalances. Experiments show that this method is effective on real-world one-dimensional datasets.

**Strengths:**

- An effort to address an exciting problem.
- A two-level approach to deep imbalanced regression.

**Weaknesses:**

- The proposed approach's effectiveness and soundness are not well justified, mainly because there is no theoretical proof.
- The claim in lines 60-62 states, "Therefore, semantic similarity, another aspect of the continuity in DIR where the similarity across labels would also reflect the similarity of their features, is **always** overlooked by the previous works." is not entirely valid, since there are several approaches modelling continuity in the feature spaces [1,2,3].
- The extension of the proposed method to multi-dimensional label spaces has not been investigated or experimented with.
 - The discussion in section 2.2 is not convincing of how semantic similarity is modelled with the proposed approach.
- The ablation study is not complete.

**Questions:**

- How this method is compared to [4]?
- How is LDS in 3.3 different from [1]?
- How  $g_{hard}$ is selected?





[1] Delving into Deep Imbalanced Regression,  ICML 2021.

[2] RankSim: Ranking Similarity Regularization for Deep Imbalanced Regression, ICML 2022.

[3] ConR: Contrastive regularizer for deep imbalanced regression, ICLR 2024.

[4] Group-aware Contrastive Regression for Action Quality Assessment, ICCV 2021.

---

### Meta-Review · Area_Chair_Q9G1 · 2024-12-19

**Metareview:**

The paper proposes an end-to-end solution for Deep Imbalanced Regression (DIR) by integrating classification regularization through asymmetric soft labeling and instance label distribution smoothing. However, the overall review is negative, with reviewers expressing concerns about the unclear novelty of the approach and the lack of experiments involving multi-dimensional label spaces. Based on these considerations, the paper is not recommended for acceptance at this time.

**Additional Comments On Reviewer Discussion:**

During the rebuttal period, the reviewers' opinions remained unchanged.

---

### Decision · Program_Chairs · 2025-01-22

Reject